# Balanced Definition of Thresholds for Mode Tracking in a Long-Term Seismic Monitoring System

Stefania Coccimiglio *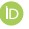, Gaetano Miraglia *, Giorgia Coletta, Rodolfo Epicoco and Rosario Ceravolo

Department of Structural, Geotechnical and Building Engineering (DISEG), Politecnico di Torino, 10129 Torino, Italy; giorgia.coletta@polito.it (G.C.); s188231@studenti.polito.it (R.E.); rosario.ceravolo@polito.it (R.C.)
* Correspondence: stefania.coccimiglio@polito.it (S.C.); gaetano.miraglia@polito.it (G.M.)

**Abstract:** The catastrophic events of recent years have strengthened the awareness of the fragility of the built heritage and the importance of careful and targeted maintenance. This, in combination with the development of modern techniques for the analysis of large datasets, has favoured the diffusion of long-term seismic monitoring systems for the protection of structures. In the field of structural health monitoring, data-driven techniques allow crucial information to be extracted from measurements without the need to model the physical phenomena involved, circumventing potential limitations that may arise. On the other hand, however, the results of data-driven approaches are based entirely on the measured structural response; this is why a high reliability of the procedure for extracting diagnostic parameters is essential. In this perspective, a Mode Tracking procedure is proposed to obtain coherent time histories of the modal frequencies of a structure as environmental conditions vary. The procedure is applied to the Sanctuary of Vicoforte, an important monumental structure located in Piedmont, known for its imposing oval dome and characterized by a permanent structural monitoring system. This study aims to disentangle the frequency time series and obtain a rigorous database on which to set up damage identification processes.

**Keywords:** Structural Health Monitoring; natural frequencies; mode tracking; permanent monitoring systems; seismic monitoring

## 1. Introduction

A form of protection of civil structures that has become increasingly widespread in recent years is represented by Structural Health Monitoring (SHM). Ambitious infrastructures or historical buildings considered fragile are advantageous targets of SHM procedures that allow their evolving conditions to be analysed, with the aim of intercepting any damage and degenerative processes early and, therefore, promptly implementing protective measures and interventions [1–7]. When studying the health state of structures, a key concept is represented by the meaning of damage state. The structure is defined as "damaged" when, although it can still work satisfactorily and safely, it no longer performs in its ideal condition, configuring a non-optimal situation [8–11]. This definition implies that the analysis of the damage state of a structure is based on the comparison between two different states of the system, one representing the initial condition and the other a second state in which the potential onset of damage is studied. The process of damage identification in structural and mechanical systems focuses precisely on this problem [12,13]. In order to evaluate the state of a structure and, therefore, a possible onset of damage, it is necessary to identify diagnostic parameters [14] that depend on the state of health of the structure, such as the natural frequencies and the modes of the system, for example. Data-driven approaches in SHM are generally used in the case of data coming from seismic permanent or long-term monitoring systems installed on buildings since a lot of samples are available to set up a reliable damage detection procedure [15] without the need to conceive a model of the

problem. These kinds of data can be characterized by trends triggered by changes in the external environment or by noise, which must be isolated and removed in order to be able to evaluate the actual health state without confounding effects [16–18]. As a matter of fact, processing errors can lead to a misinterpretation of the path of natural frequencies when monitored over time, leading to incorrect diagnoses. This is the reason why the dynamic identification process, including the phase of connecting the results of each acquisition (which, in the case of permanent monitoring systems, is generally automated), must be carefully conceived and calibrated to the specific needs of the observed object.

Connecting the results of different acquisitions means tracing the vibration modes over time, associating each identified mode to the previously identified ones, building step by step well separated time series of frequencies (Mode Tracking problem). The importance of this phase is linked to the fact that in the context of dynamic monitoring, the diagnosis of a structure is not based on the absolute values of the parameters obtained (i.e., it cannot be said whether a structure is healthy or not based on its one-time obtained frequencies), but rather on a comparison established between the data acquired at a time when the structure is considered healthy and all subsequent data. This comparison must therefore be consistent, providing for the juxtaposition of the different quantities (frequency, shape, and modal damping) identified over time, always relating to the same vibration mode. In fact, if values relating to two different modes are associated and considered as if they were the same, the errors could be interpreted by the damage detection algorithms as the onset of anomaly.

The most immediate strategy to conceive for carrying out Mode Tracking is to rely on a similarity parameter between modes, for example the Modal Assurance Criterion (MAC), and associate the modes that present the greatest value, beyond a certain minimum threshold. However, considering that not all identifications contain the same modes of vibration (especially in Operational Modal Analysis, OMA, where environmental excitation may not amplify all modes, making some difficult to identify) and that spurious modes may result in automatic identification procedures, this strategy could run into many wrong associations. Furthermore, since higher frequency modes are generally more difficult to identify and sometimes less accurate, defining a single MAC threshold for all modes could lead to inappropriately discarding a lot of useful data. The same thing, even on the main vibration modes, could happen if the installed system is sparse (few sensors), as the mode coupling errors rise drastically due to the drastic loss of robustness of the MAC, the parameter used to correlate the different modes of vibration. Discarding so much data leads to a lack of timely monitoring, which could instead cause vital information to go unnoticed.

For all these reasons, in this paper, a procedure to ensure correct tracking of natural frequencies of structures is presented, which seeks to remedy the drawbacks listed above. The data used comes from the seismic permanent monitoring system installed on the Sanctuary of Vicoforte, a monumental church located in the northwest of Italy. This paper is divided as follows: in Section 2, the characteristics of long-term dynamic monitoring systems are exposed, and a focus is made on the case study of the Sanctuary of Vicoforte; then, in Section 3, the tracking procedure is explained in general terms, while in Section 4, the specific procedure is presented together with the obtained results. In Section 5, the results and discussion of the analysis are reported. Finally, the conclusions are illustrated in Section 6, together with the problems encountered in the applied procedure and future developments.

## 2. Long-Term Seismic Monitoring Program

Since damage identification is determined by changes in the dynamic response of systems [12], a meaningful step of the SHM process is the extraction of dynamic features of the structures, i.e., natural frequencies and modal shapes. Natural frequencies are obtained from the signals recorded by in situ sensors (e.g., accelerometers) of dynamic monitoring systems through system identification tasks [19]. In general, when dealing with structural monitoring, it is useful to combine information from static structural systems with dynamic ones to get a comprehensive view of the health state of the structure. This is because

dynamic monitoring aims to provide parameters that characterize the structure from a global point of view. There are different types of dynamic monitoring systems: one-time monitoring, periodic, and continuous (or permanent) monitoring. In the first two cases, the instruments are used to perform dynamic tests by measuring characteristic vibrations of the structure induced by external forces or natural phenomena, and they are removed after obtaining a predetermined amount of data. In the case of continuous monitoring, the sensors are fixed to the structure permanently, and they record the vibrations of the structure whenever a microquake occurs or a significant source of vibration exceeds a certain threshold (trigger-based monitoring). They are also capable of recording the vibration of the structure continuously or at short intervals of time under environmental conditions. In this case, it is referred to as Vibration Based SHM (VB-SHM) [19–21]. Continuous dynamic monitoring, through the remote and instantaneous reading of signals, such as displacements, velocities, and accelerations, can enable real-time or near real-time knowledge of useful information about the overall health state of the structure and the onset or development of serious damage states. Therefore, once the signals have been recorded, stored, and pre-processed, the modal parameters identification, both in operating and extreme conditions, is of fundamental interest. The importance of OMA arises from the dependence of the dynamic behaviour of the structure on its intrinsic characteristics, such as mass, stiffness, damping, etc. [22]. Therefore, if no changes occur within the building (such as structural damage), the dynamic behaviour of the structure remains constant, unless the presence of operational and environmental changes; on the contrary, in the presence of damage phenomena, changes occur in the dynamic parameters of the structure. These data coming from permanent monitoring systems are applied in data-driven approaches for VB-SHM, and then statistical models of the system can be defined, and noise levels and environmental variations are established naturally [15,23].

*The Case of the Sanctuary of Vicoforte*

The Sanctuary of Vicoforte (CN) (Figure 1) is one of the main monumental structures of the Italian Baroque, whose importance is also due to its imposing dome. With a major axis of 37.23 m and a minor axis of 24.89 m, it is the largest oval masonry dome in the world [24]. More information about the Sanctuary of Vicoforte can be found in [24].

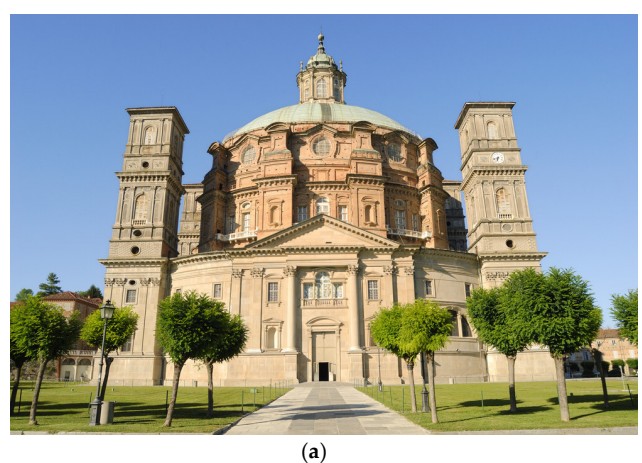
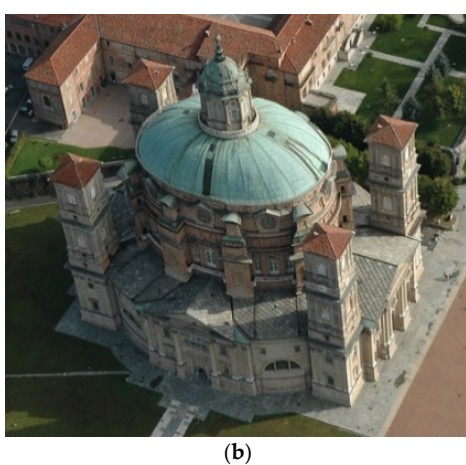

(**a**)                                                                                   (**b**)

**Figure 1.** The Sanctuary of Vicoforte: (**a**) front external view and (**b**) top external view.

Over the years, the Sanctuary of Vicoforte has been the subject of various types of investigation [24–26]. Moreover, the evolution of the damage in the sanctuary has been monitored since 1985, when the first strengthening system was installed [24]. A static sensors system was placed in 2004 and has been refurbished within the last year. Finally, since 2015, the sanctuary has been equipped with a permanent dynamic system that allows the monitoring of the health state of the structure in almost real time. In the following,

since this paper proposes a MT strategy, the details of only the dynamic system will be exposed below.

The permanent monitoring of the lantern–dome–drum system of the Sanctuary of Vicoforte consists of 12 accelerometers (PCB Piezotronics, model 393B12, seismic, high sensitivity, ceramic shear ICP® accel., 10 V/g, 0.15 to 1k Hz). Their positions are shown in Figure 2: three orthogonal accelerometers are installed at the base of the crypt in order to record the ground accelerations, and nine accelerometers are located at different heights on the lantern–dome–drum system along both longitudinal, transversal, and vertical directions. The acquisition system is designed as a master/slave scheme to limit the distortion due to cable length that becomes significant over 50 m [19]. The acquired signals are locally collected in the Sanctuary of Vicoforte and transmitted online to the Earthquake Engineering and Dynamics Laboratory (EED Lab) of the Politecnico di Torino. Once the signals of acceleration are acquired by the accelerometers, they are then used to automatically estimate the natural frequencies and mode shapes of the sanctuary. The dynamic system data is available from 2016 to now with some interruptions due to various causes such as electric issues or modifications and maintenance performed to the sanctuary and the sensing system (e.g., time lasted to wait for the arrival of burned electronic components). Figure 3 shows the days when acceleration data are available; red squares represent missing data, while green square indicates those days in which the acquisitions are available.

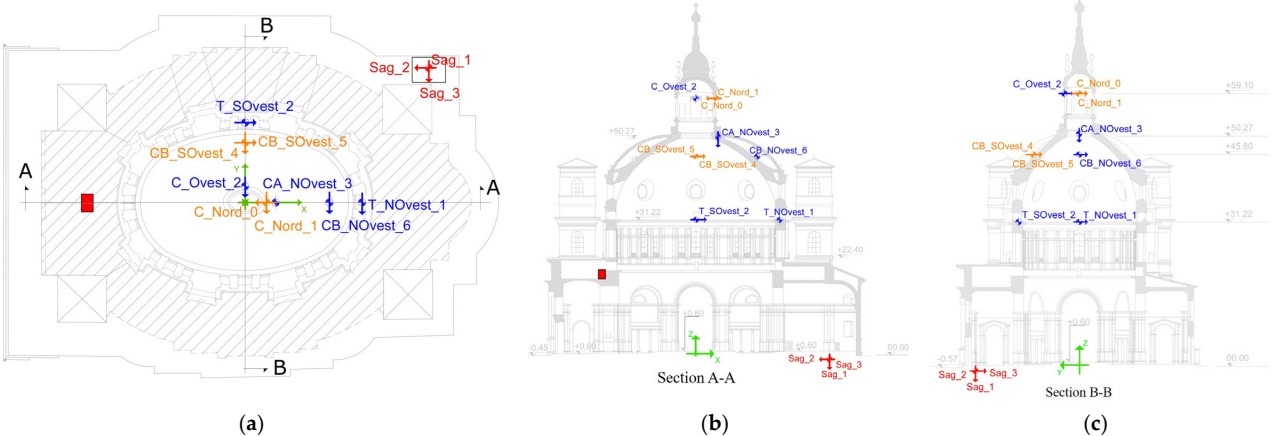

(**a**)  (**b**)  (**c**)

**Figure 2.** Permanent dynamic monitoring system: (**a**) plan, (**b**) longitudinal section, and (**c**) cross-section.

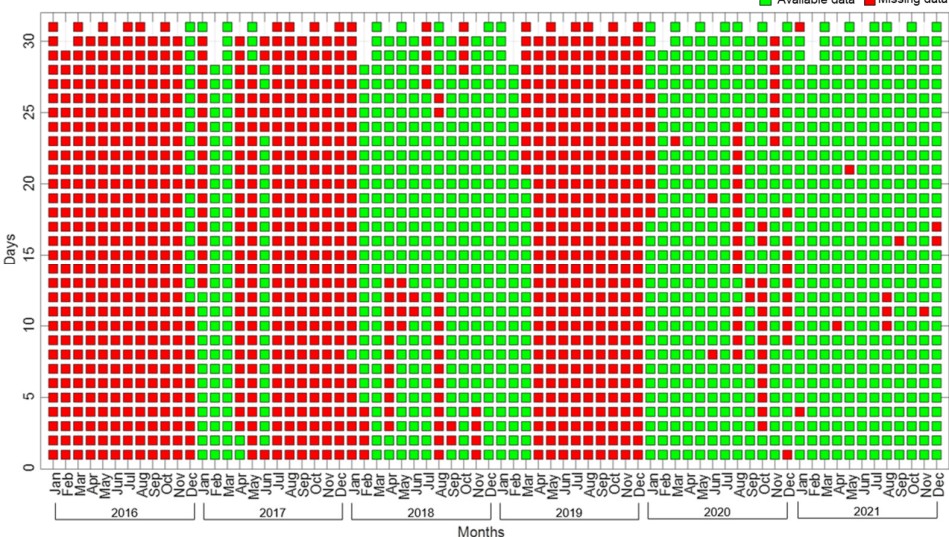

**Figure 3.** Days with available recordings from the permanent systems of the Sanctuary of Vicoforte.

## 3. Mode Tracking

Once the dynamic parameters of the structure are estimated through automated OMA techniques, an important and challenging step consists of progressively assigning the identified modal properties of different identification tasks performed over time to previously estimated vibration modes, also due to the presence of environmental and operational variations (EOVs) which do not affect the quality of the transmitted data but could affect the structural response of the building and consequently the recorded signals. This process is known as Mode (or Modal) Tracking (MT) and is commonly achieved by comparing estimated natural frequencies and their associated mode shapes with the modal properties of a set of reference modes [16]. The reference modes are obtained from an identification task in which most of the structural modes are identified or can be derived from one-time monitoring and in situ tests [27]. There are several examples where effective MT procedures are shown and experimentally validated [16,28,29]. However, the MT procedure may vary from structure to structure, especially in the case of complex civil structures, and the result may also be affected by external factors, such as environmental conditions, which can cause changes in the dynamic response of the structure. In the present application, a set of reference modes is compared with each identification task result. The comparison is performed in terms of MAC. The MAC between reference modes and modes identified in several tasks is calculated. Within the comparison of a single identification task result, the mode with maximum MAC with a specific reference mode is attributed to the time history of that reference mode. However, due to low natural excitation, the identification tasks can fail to identify all the reference modes, or the correlation (maximum MAC value) may be too low to consider a newly identified mode associated with a previous reference time history. For this reason, a threshold on the minimum value of acceptable maximum MAC is imposed, under which the mode remains unclassified, and the time history is filled up with no number, i.e., Not a Number (NaN) value. In detail, for a specific reference mode, the maximum MAC is calculated as follows:

- Take a time instant $t_i$;
- For each time instant, a set of eigenvectors is identified;
- Calculate the $MAC_k$ between the eigenvector of a specific reference mode and the $k_{th}$ eigenvector identified for each time instant $t_i$;
- For each time instant $t_i$, a set of MAC can be calculated;
- For each time instant $t_i$, take the maximum among the several MAC and get: $MAC_{max,i} = max_k[MAC_k]$;
- If $MAC_{max,i} < MAC_{thr}$, fill the time history of the natural frequency at $t_i$ with NaN; otherwise, the frequency at $t_i$ is the frequency $f_{k^*}$ related to $MAC_{max,i} = MAC_{k^*}$.

## 4. Analysis

### 4.1. Definition of Thresholds for Mode Tracking

In the case of the Sanctuary of Vicoforte, the modal parameters of the structure are estimated through an automatic modal identification procedure fully described in [19]. Once this procedure was applied for the entire acquisition period (Figure 3), it was possible to obtain the time series of the natural frequencies and their mode shapes through the MT procedure. In this study, to evaluate the coupling between the reference modes and the remaining vibration modes obtained from the identification process, MAC was used [30]. In particular, the analysis was performed through a balanced definition of MAC threshold ($MAC_{thr}$) values, which have been chosen starting from experimental data in order to have the highest autocorrelation and minimum loss of information of the resulting time histories of natural frequencies. This allowed the efficient tracking of the natural frequencies. $MAC_{thr}$ was the only parameter changed during the analysis, and it was used to pass from time-uncorrelated natural frequencies to autocorrelated time histories of the natural frequencies. The threshold on the error in mode correlation, $e_{thr}$, was evaluated with Equation (1).

$$e_{thr} = 1 - MAC_{thr} \qquad (1)$$

A $MAC_{thr}$ equal to 0.95 was considered as the starting point because this value was used in the stabilization process. Then, this value was changed, considering the following aspects:

- Outlier analysis;
- Total number of missing values, i.e., NaN;
- Problem of classification, this means that there are many trends at different frequencies rather than just one that oscillates within small ranges.

Thus, the choice of the threshold value considered the reduction in the number of outliers, also considering the Standard Deviation (StD), $\sigma$, of the time series obtained, as reported in Equation (2).

$$\sigma = \sqrt{\frac{1}{N-1}\sum_{i=1}^{N}|x_i - \mu|^2} \qquad (2)$$

$N$ is the number of samples, $x$ is the observation, and $\mu$ is the mean of $x_i$; see Equation (3).

$$\mu = \frac{1}{N}\sum_{i=1}^{N}x_i \qquad (3)$$

At the same time, as the outliers decreased, the increase in missing values, $m_v$, was taken into account so as not to lose any useful information (Equation (4)).

$$N_{NaN} = \sum_{i=1}^{N}m_v \qquad (4)$$

where $N_{NaN}$ is the total number of missing values. In addition, the results of the latter may depend on several factors, such as the number of data available from the identification process or the reference mode chosen. Regarding the latter, it can be characterized by different frequency values and mode shapes, which are influenced by environmental factors such as temperature [15]. For this reason, in this study, it has been decided to take as reference three specific moments referring to distant periods of the year corresponding to different seasons: the first reference set of modes in February (low temperature, Section 4.2), the second in May (mild temperature, Section 4.3), and finally the third in August (high temperature, Section 4.4). In all cases, the first seven frequencies were analysed as they can be considered the most stable; indeed, they are characterized by higher identification percentages. Therefore, the following modes were evaluated: the first and second transversal ($f_1$ and $f_6$), the first and second longitudinal ($f_2$ and $f_7$), the first torsional ($f_3$), and the first and second local modes of the dome (i.e., ovalization) ($f_4$ and $f_5$).

*4.2. Low Temperature Case*

The low temperature analysis takes the identification of 5 February 2018 as the reference set of modes. Since the analysis involves the first seven modes, that observation can be considered an appropriate choice because twelve modes have been correctly identified in it.

The first step of the analysis was the choice of the $MAC_{thr}$ according to a trial-and-error procedure that took into account the aspects listed above. Therefore, the threshold value was calibrated from the starting point of 0.95 and increased (Figure 4). The procedure was first performed on $f_1$, $f_2$, and $f_3$ and then extended to $f_4$, $f_5$, $f_6$, and $f_7$. By increasing the value from 0.9500 to 0.9899, it is possible to observe a reduction in outliers and Standard Deviation; however, from a certain point onwards, there could also be a loss of meaningful information since the number of NaN values in the natural frequencies increases. Figure 5 shows the trend of Standard Deviation and missing values as the $MAC_{thr}$ changes. In all three cases ($f_1$, $f_2$, and $f_3$), StD decreases while $N_{NaN}$ increases. Based on this and referring to the third frequency, a $MAC_{thr} = 0.9750$ was chosen as the optimal value. The second step consists of obtaining the tracking of the remaining frequencies using the $MAC_{thr}$; however,

by carrying out an analysis taking into account all seven frequencies with a threshold of 0.9750, new outliers are introduced with values very far from the mean value.

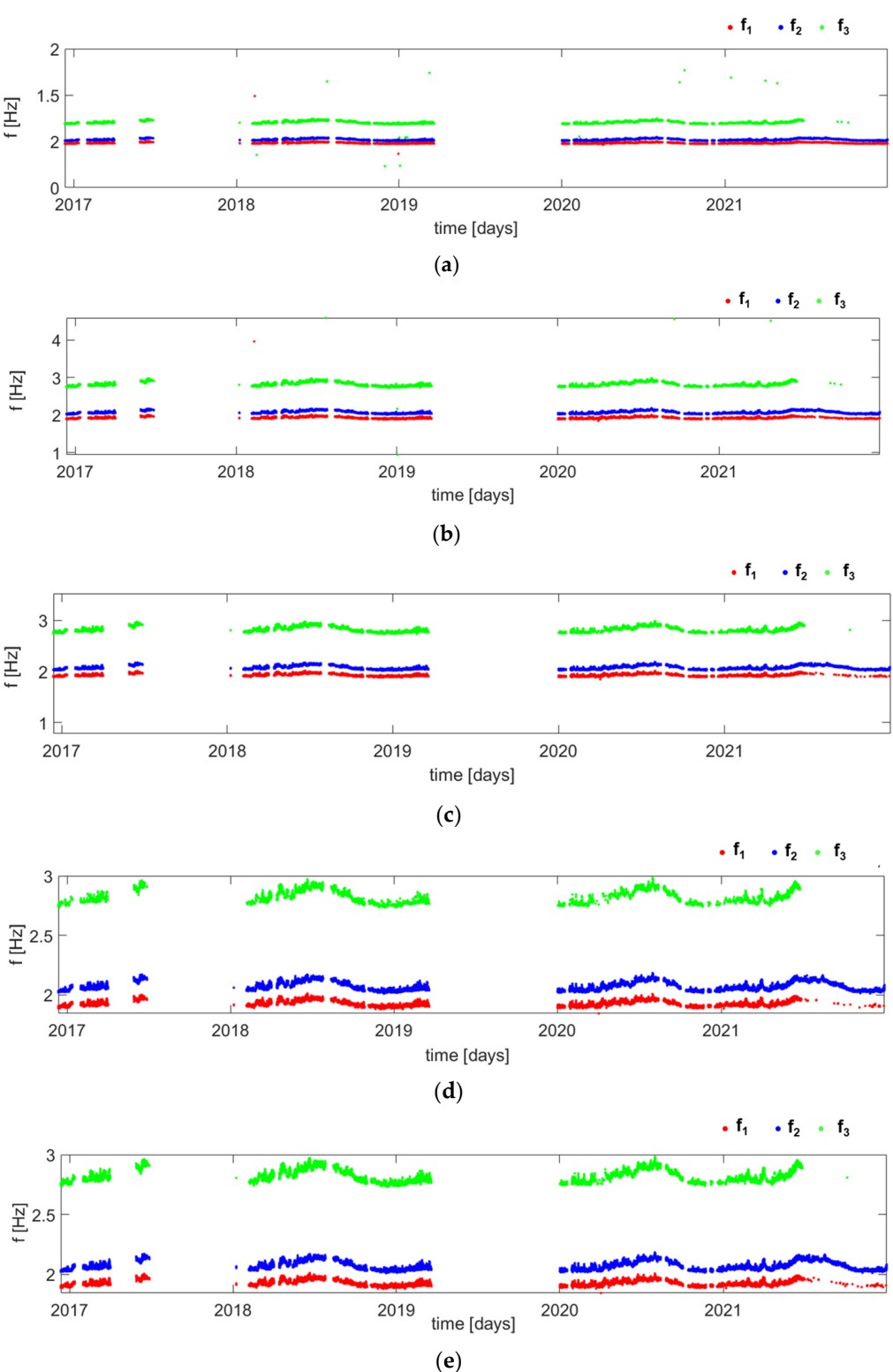

**Figure 4.** Comparisons of $f_1$, $f_2$, and $f_3$ based on different values of MAC threshold: (**a**) $MAC_{thr} = 0.9500$, (**b**) $MAC_{thr} = 0.9600$, (**c**) $MAC_{thr} = 0.9750$, (**d**) $MAC_{thr} = 0.9899$, and (**e**) $MAC_{thr} = 0.9800$.

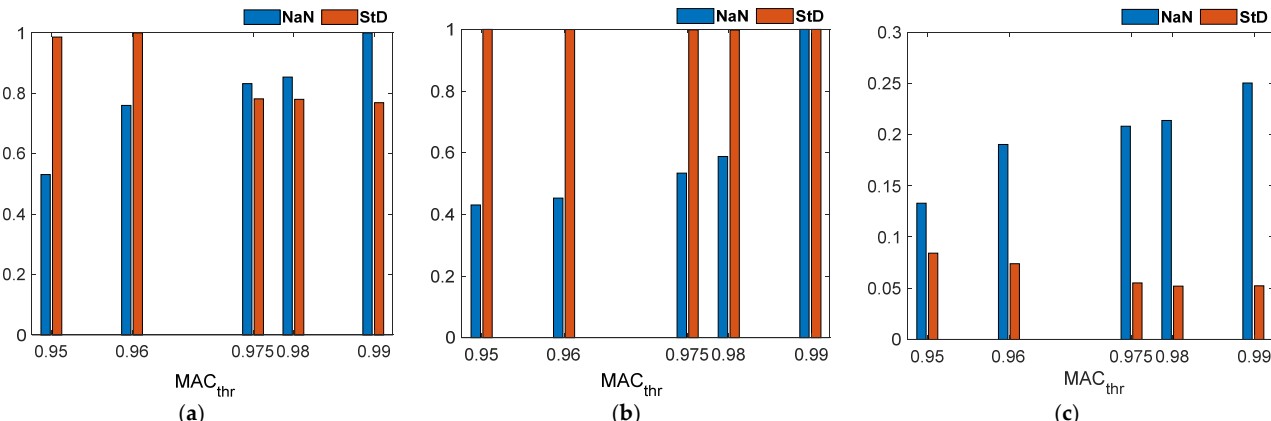

**Figure 5.** Number of missing values (NaN) and Standard deviation for different MAC thresholds (normalized values): (**a**) $f_1$, (**b**) $f_2$, and (**c**) $f_3$.

These results are shown in Figures 6 and 7. Since this procedure causes a large variation in the time series of the first three frequencies, it was decided to proceed by analysing the frequencies in blocks, i.e., $f_1$–$f_3$, $f_4$–$f_5$, and $f_6$–$f_7$, and finding a different threshold for each of these blocks.

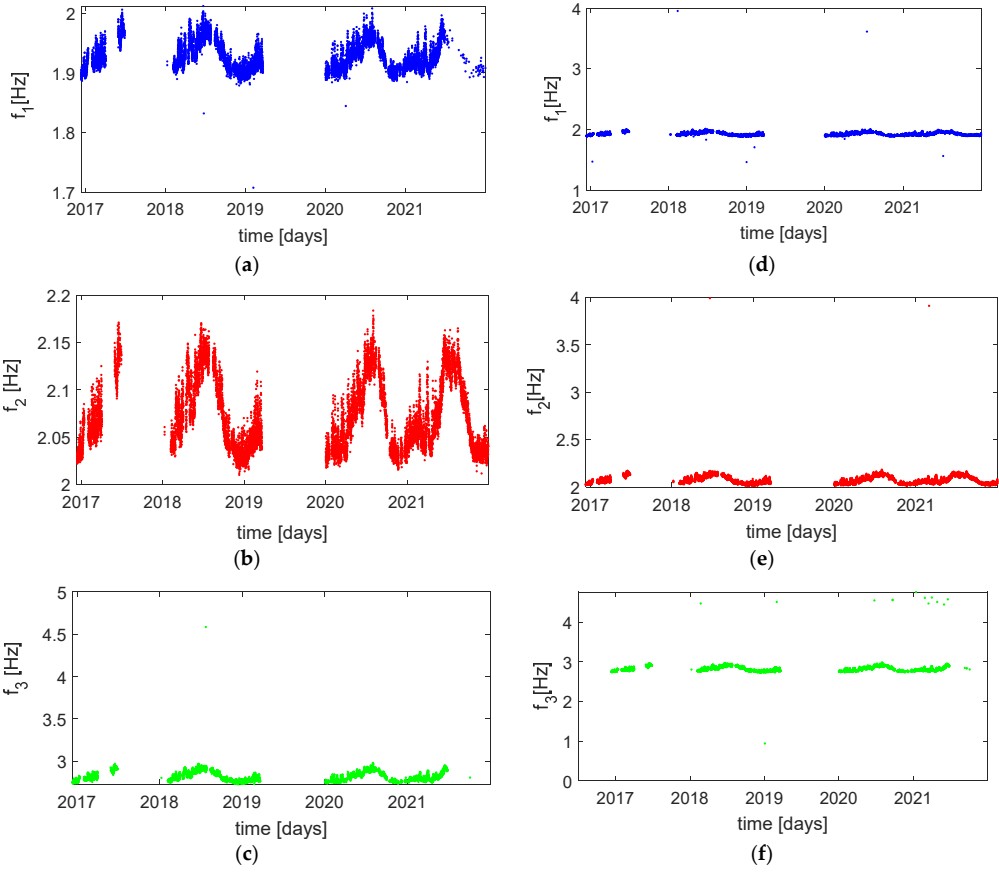

**Figure 6.** Comparison between different MT approaches with a MAC threshold of 0.9750: (**a**) $f_1$ time series considering only the first three frequencies, (**b**) $f_2$ time series considering only the first three frequencies, (**c**) $f_3$ time series considering only the first three frequencies, (**d**) $f_1$ time series considering seven frequencies ($f_1$–$f_7$), (**e**) $f_2$ time series considering seven frequencies ($f_1$–$f_7$), and (**f**) $f_3$ time series considering seven frequencies ($f_1$–$f_7$).

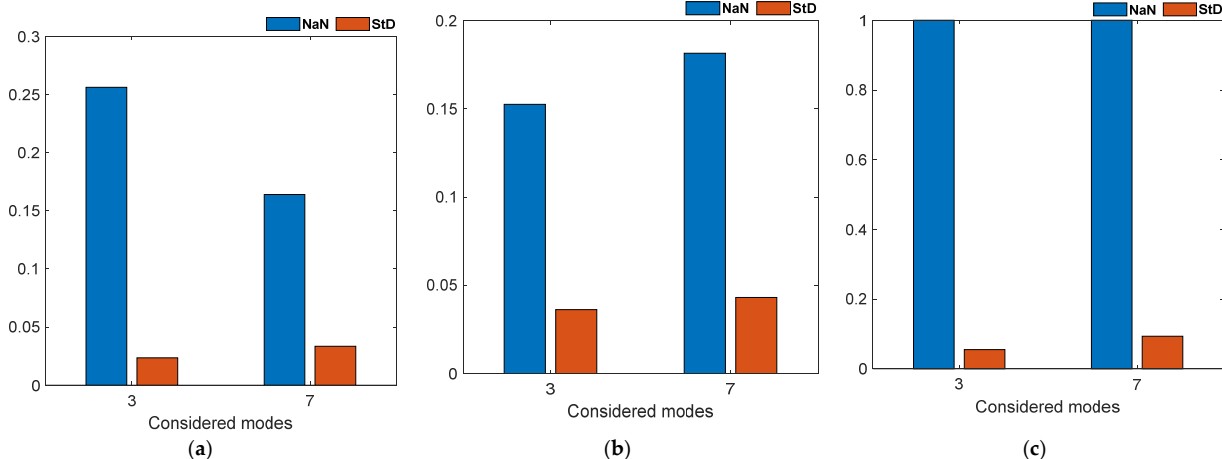

**Figure 7.** Comparisons of number of missing values (NaN) and Standard Deviation (StD) considering only the first three frequencies with a MAC threshold of 0.9750 and considering seven frequencies ($f_1$–$f_7$) with a MAC threshold of 0.9750: (**a**) $f_1$, (**b**) $f_2$, and (**c**) $f_3$.

Once the same procedure of the $f_1$–$f_3$ block was adopted, an optimal value of $MAC_{thr} = 0.9890$ was found in the case of $f_4$–$f_5$ because by increasing the value, the frequency tracking does not vary considerably, i.e., neither Standard Deviation nor number of missing values change. By further increasing this value and reaching up to approximately 0.9930, the outliers decrease markedly; however, together with them, a lot of data is lost, and consequently, it is necessary to carefully evaluate whether the reduction in the Standard Deviation is more convenient than a significant loss of data. Finally, in the case of $f_6$–$f_7$, a $MAC_{thr} = 0.9892$ was chosen. The results are shown in Figure 8, while the tables show the different values of MAC threshold considered in both cases of $f_4$–$f_5$ and $f_6$–$f_7$ and the relative number of missing values and Standard Deviation (Tables 1 and 2); the final chosen values have been highlighted.

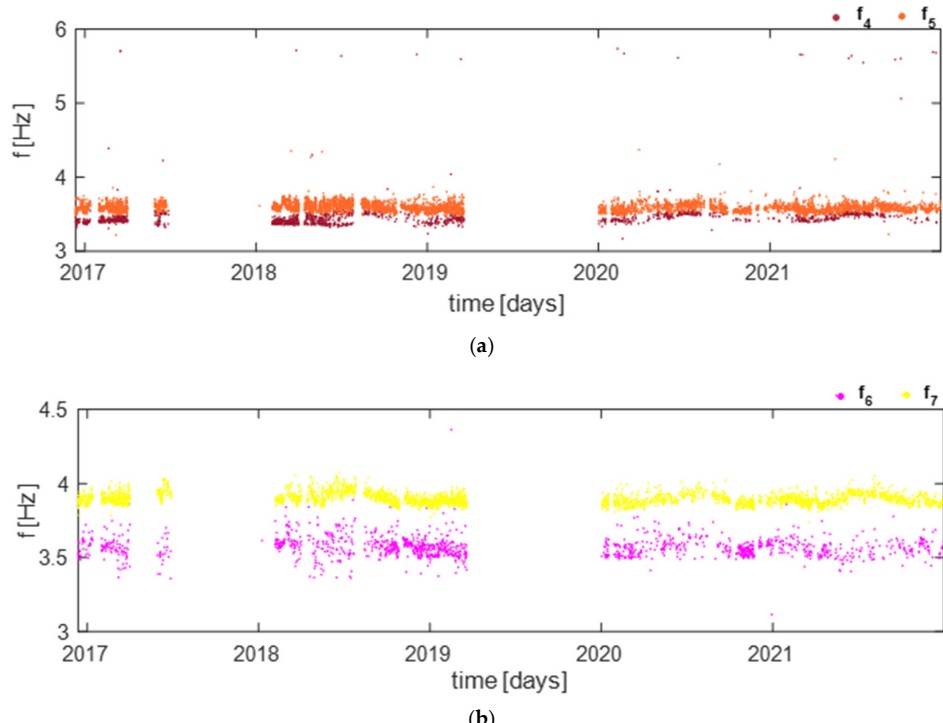

**Figure 8.** Time series of frequencies: (**a**) time series of $f_4$ and $f_5$ with a MAC threshold of 0.9890 and (**b**) time series of $f_6$ and $f_7$ with a MAC threshold of 0.9892.

**Table 1.** Summary table with the MAC threshold values for $f_4$ and $f_5$ in the case of low temperature.

|  | $MAC_{thr}$ | NaN | StD |
|---|---|---|---|
| | 0.9750 | 7569 | 0.2789 |
| | 0.9850 | 11,275 | 0.2401 |
| $f_4$ | 0.9877 | 12,370 | 0.2531 |
| | 0.9890 | 12,856 | 0.2458 |
| | 0.9930 | 13,919 | 0.1178 |
| | 0.9750 | 8539 | 0.0890 |
| | 0.9850 | 8718 | 0.0639 |
| $f_5$ | 0.9877 | 8829 | 0.0594 |
| | 0.9890 | 8895 | 0.0587 |
| | 0.9930 | 9364 | 0.0546 |

**Table 2.** Summary table with the MAC threshold values for $f_6$ and $f_7$ in the case of low temperature.

|  | $MAC_{thr}$ | NaN | StD |
|---|---|---|---|
| | 0.9750 | 3484 | 0.0908 |
| | 0.9850 | 9495 | 0.0985 |
| $f_6$ | 0.9877 | 11,989 | 0.0924 |
| | 0.9892 | 13,059 | 0.0697 |
| | 0.9940 | 14,387 | 0.1246 |
| | 0.9750 | 8229 | 0.0405 |
| | 0.9850 | 9862 | 0.0384 |
| $f_7$ | 0.9877 | 10,612 | 0.0379 |
| | 0.9892 | 11,159 | 0.0379 |
| | 0.9940 | 13,260 | 0.0360 |

*4.3. Mild Temperature Case*

In the second case, the results of the identification of 15 May 2018 are considered as reference mode. Also, in this case, a threshold of 0.95 was taken as the starting value. For the first three frequencies, it was possible to consider the same value used in the previous case (Section 4.2), i.e., 0.9750, since, in addition to giving rise to excellent results, it is preferable to keep this parameter constant for uniformity, whenever possible (Figure 9a). To obtain the trend of $f_4$ and $f_5$, the analysis started from the value of $MAC_{thr}$ = 0.989 used in the case of low temperature. In this case, however, this value is not adequate as both frequencies are characterized by many outliers, which do not allow for a well-defined trend. As a subsequent step, a lower value of 0.9777 was considered, and a worsening was found, as in the case of $f_5$, the outliers increased, while for $f_4$, there was a classification problem that generated multiple behaviours within the same frequency. For this reason, it was decided to test a value higher than 0.989, and therefore, a MAC threshold equal to 0.9920 was considered. In this case, the tracking improves as the trend of both frequencies begins to be cleaner; however, various outliers are present. Further increasing and testing different values, the best result is given by 0.9955 as many outliers are eliminated, and there is not much data loss (Figure 9b). Even in the case of $f_6$ and $f_7$, the starting value of low temperature was used, i.e., 0.9892, and some difficulties were encountered in finding a single value that would make tracking acceptable in the case of both frequencies. For this reason, it has been decided to analyse the two frequencies independently; but even in this case, satisfactory results were not achieved, consequently the first type of results were maintained with a $MAC_{thr}$ = 0.9800 (Figure 9c). Table 3 shows the different values of the MAC threshold considered in the case of $f_4$–$f_7$ and the relative number of missing values and Standard Deviation.

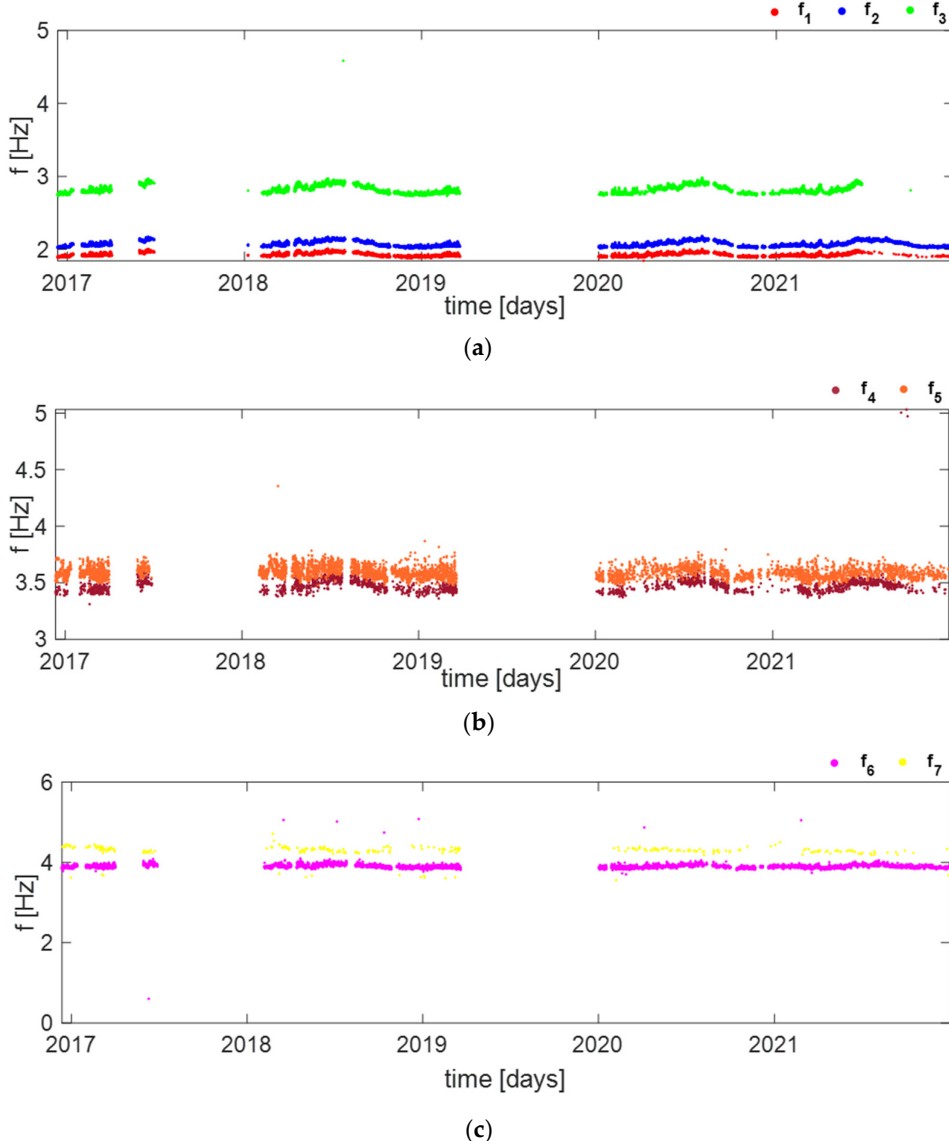

**Figure 9.** Time series of frequencies: (**a**) time series of $f_1$, $f_2$, and $f_3$ with a MAC threshold of 0.9750, (**b**) time series of $f_4$ and $f_5$ with a MAC threshold of 0.9955, and (**c**) time series of $f_6$ and $f_7$ with a MAC threshold of 0.9800.

*4.4. High Temperature Case*

The high temperature analysis takes the identification of 14 August 2018 as the reference mode. Also, in this case, the $MAC_{thr}$ value used for low and mild temperatures works well for the first three frequencies; therefore, in the case of $f_1$–$f_3$, the value of 0.9750 was maintained. However, for all the other frequencies considered, the analysis becomes more complicated because they are characterized by more classification errors (i.e., number of classes or components of a Probability Distribution Function fitted to a tentative time history of the modal parameters), consequently, isolating a defined behaviour that represents the actual tracking is really challenging. In this case, given the large presence of outliers, instead of starting from the value of 0.95, the decision was made to start from the maximum value of 0.99 and then decrease to observe the progressive restocking of the time series. Furthermore, MT was performed independently for $f_4$ to $f_7$. For $f_4$, a value of 0.9930 was chosen, and an acceptable trend was obtained; even in the case of $f_6$, it was possible to define a MAC threshold value that allowed to trace the time series, and this is equal to 0.9701. For $f_5$ and $f_7$, it was not possible to find a value of the MAC threshold because, in the first case, a lot of data is lost, so it is not possible to have a defined trend since it seems

to be different behaviours at different frequencies; in the second case, the main problem is due to the presence of two different trend around 3 and 4.5 Hz that make the tracking result unacceptable. For these last reasons, for high temperatures, only the time series of $f_1$–$f_4$ and $f_6$ are shown Figure 10 and Table 4.

**Table 3.** Summary table with the MAC threshold values for $f_4$–$f_7$ in the case of mild temperature.

|  | **MAC$_{thr}$** | **NaN** | **StD** |
|---|---|---|---|
| **$f_4$** | 0.9777 | 5862 | 0.2177 |
|  | 0.9890 | 9496 | 0.1247 |
|  | 0.9920 | 10,558 | 0.1006 |
|  | 0.9955 | 12,363 | 0.0723 |
|  | 0.996 | 12,683 | 0.0667 |
| **$f_5$** | 0.9777 | 8356 | 0.0807 |
|  | 0.9890 | 8668 | 0.0568 |
|  | 0.9920 | 8968 | 0.0544 |
|  | 0.9955 | 10,040 | 0.0515 |
|  | 0.9960 | 10,341 | 0.0495 |
| **$f_6$** | 0.9700 | 13,704 | 0.2252 |
|  | 0.9750 | 13,935 | 0.1674 |
|  | 0.9780 | 14,068 | 0.1618 |
|  | 0.9800 | 14,149 | 0.1497 |
|  | 0.9892 | 14,440 | 0.2269 |
| **$f_7$** | 0.9700 | 8167 | 0.1274 |
|  | 0.9750 | 8825 | 0.0915 |
|  | 0.9780 | 9329 | 0.0746 |
|  | 0.9800 | 9732 | 0.0729 |
|  | 0.9892 | 12,283 | 0.0378 |

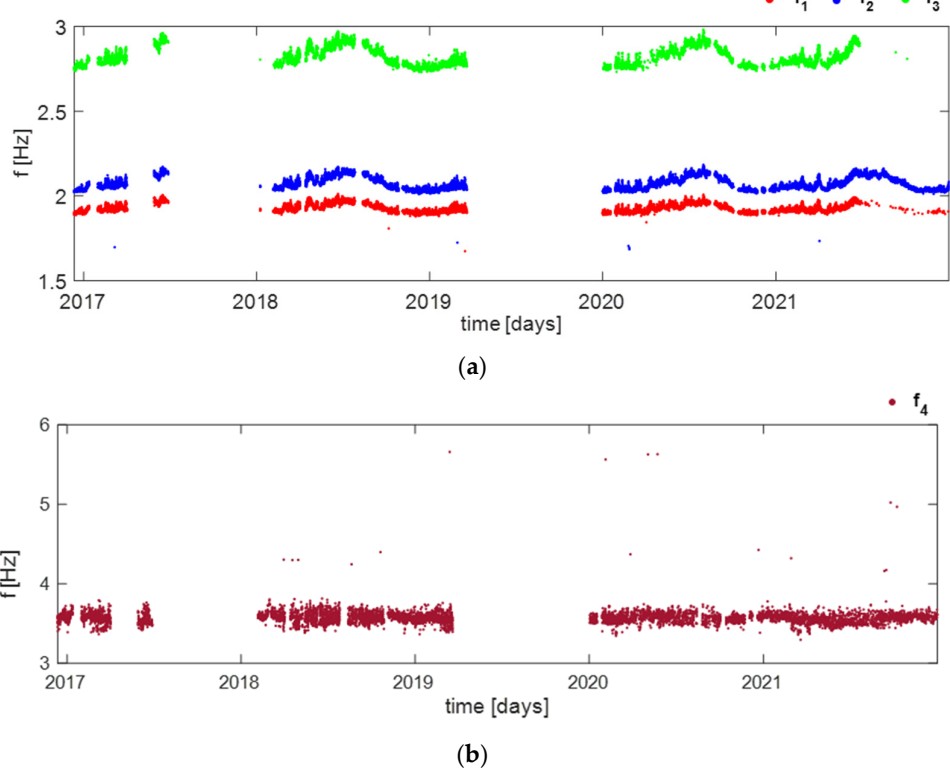

**Figure 10.** *Cont.*

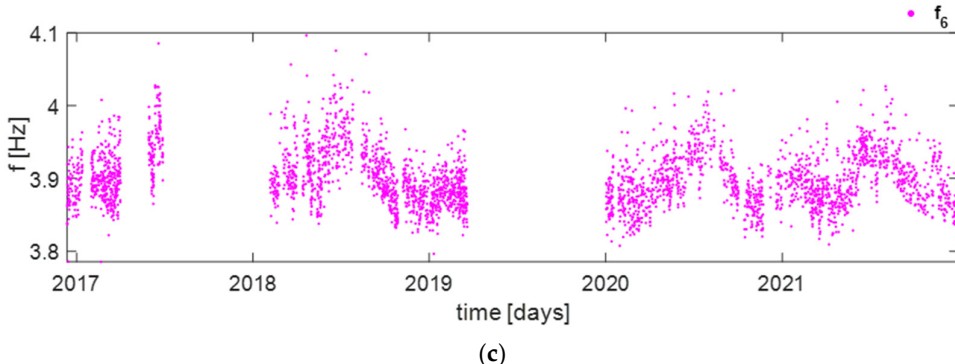

(c)

**Figure 10.** (a) Time series of $f_1$, $f_2$, and $f_3$ with a MAC threshold of 0.9750, (b) time series of $f_4$ with a MAC threshold of 0.9930, and (c) time series of $f_6$ with a MAC threshold of 0.9701.

**Table 4.** Summary table with the MAC threshold values for $f_4$ and $f_6$ in the case of high temperature.

| | $MAC_{thr}$ | NaN | StD |
|---|---|---|---|
| | 0.9650 | 2137 | 0.1097 |
| | 0.9700 | 3116 | 0.0951 |
| $f_4$ | 0.9930 | 11,372 | 0.0658 |
| | 0.9950 | 12,724 | 0.0498 |
| | 0.9988 | 14,387 | 0.0450 |
| | 0.9700 | 11,132 | 0.0418 |
| | 0.9701 | 11,149 | 0.0383 |
| $f_6$ | 0.9705 | 11,222 | 0.0382 |
| | 0.9740 | 11,755 | 0.0377 |
| | 0.9750 | 11,897 | 0.0376 |

## 5. Results and Discussion

In the case of low temperature (Section 4.2), it was possible to extract the tracking of frequencies for all considered modes, i.e., from $f_1$ to $f_7$. From the results obtained, it can be seen that tracking is more satisfactory in the case of lower and more stable frequencies; in fact, there is a lower number of NaN and less oscillation around the mean value. Secondly, for mild temperature (Section 4.3), even if it is possible to track all seven first frequencies, in the case of the higher modes, the results obtained cannot be considered satisfactory as in the case of low reference temperatures since it is not possible to define a rather "clean" trend. Finally, in the last case (Section 4.4), with the exception of the first easily obtainable frequencies, in the rest of the cases, it is very hard to clean the trend; for this reason, only some of the time series have been reported. For the first three frequencies, it could be said that they are less sensitive to the choice of the reference mode; in fact, from Figures 4, 9a and 10a and the below Table 5, it can be seen that the mean and Standard Deviation are almost unchanged, the only variation occurs in the number of missing value.

**Table 5.** Mean values, standard deviations (StD), and missing values (NaN) for different temperatures for the first three frequencies.

| | | Low T | Mild T | High T |
|---|---|---|---|---|
| | μ | 1.926 | 1.926 | 1.926 |
| $f_1$ | StD | 0.0235 | 0.0234 | 0.0236 |
| | $N_{nan}$ | 3006 | 2958 | 2949 |
| | μ | 2.072 | 2.072 | 2.073 |
| $f_2$ | StD | 0.0362 | 0.0362 | 0.0376 |
| | $N_{nan}$ | 1791 | 1773 | 3108 |
| | μ | 2.826 | 2.824 | 2.823 |
| $f_3$ | StD | 0.0551 | 0.0592 | 0.0544 |
| | $N_{nan}$ | 11,734 | 8268 | 8353 |

## 6. Conclusions

A procedure of Mode Tracking is presented in order to obtain the time series of frequencies identified from the data collected by the seismic long-term monitoring system installed on the Sanctuary of Vicoforte. To achieve the aim of this study, it was decided to use the MAC threshold as a parameter for frequency tracking. Furthermore, given the influence that environmental factors, such as temperature, can have on frequencies, cases at different temperatures were analysed by choosing three reference identifications corresponding to as many different periods of the year. In this way, it was possible to analyse the trend at low, mild, and high temperatures. From the analysis performed in this study, it is possible to state that the MT procedure cannot be defined univocally for all structures and modes. In the case of the first three frequencies, which can be defined as the most stable, the results of the threshold definition remained unchanged, and in fact, it was possible to use the same MAC threshold value for the three reference periods. For these frequencies, the trends obtained are almost similar, as demonstrated by the mean, Standard Deviation and missing values, which are almost constant. On the contrary, in the case of the higher and less stable modes, it was necessary not only to vary the threshold MAC value but also to change and adapt the procedure used based on the single case; despite this, in the case of mild temperature, the times series of higher modes are characterized by a significant number of missing value, while for high temperature, it was not possible to obtain the time series for the fifth and seventh frequencies.

From this application, it appeared that the choice of parameter to obtain mode tracking, in this case, the MAC threshold, should be calibrated case by case, both based on the different systems analysed and also based on the mode analysed within it. The MT procedure could be clearer and more immediate for structures with less complex dynamic behaviour, given by more uniform materials and simpler structural schemes, while more difficulties could arise in the case of complex ancient monumental buildings such as the Sanctuary of Vicoforte. Another source of complexity is certainly brought by the overlap of frequency ranges between the different modes, which would complicate obtaining correct tracking. In addition, there are cases where operational and environmental variations strongly influence the modal properties of the structure, so it should be necessary to take into account the variability due to these external conditions to obtain a correct estimate of the thresholds. For these reasons, automating the calibration of the MAC threshold parameter could be very challenging. In fact, even if an automated algorithm were developed to take into account the aspects listed in Section 4, defining the exact criteria for obtaining the proper threshold value would still not be easy. A final check by an expert would always be preferable and sometimes necessary.

Finally, from the analyses obtained in this study, the results of the low-temperature case have been considered, i.e., those considered most reliable, to carry out simulations capable of providing more information on the relationship between the environmental temperature and the mechanical properties of the system; this topic will be the subject of future studies.

**Author Contributions:** Conceptualization, S.C., G.M., G.C., R.E. and R.C.; methodology, S.C. and G.M.; software, S.C. and G.M.; validation, S.C. and G.M.; formal analysis, S.C. and G.M.; investigation, S.C. and G.M.; resources, S.C. and G.M.; data curation, S.C., G.M., G.C. and R.E.; writing—original draft preparation, S.C.; writing—review and editing, S.C., G.M., R.C., G.C. and R.E. visualization, S.C., G.M. and R.C.; supervision, R.C.; project administration, R.C.; funding acquisition, R.C. All authors have read and agreed to the published version of the manuscript.

**Funding:** This research received no external funding.

**Acknowledgments:** This research was partially supported by the Amministrazione del Santuario di Vicoforte and the Fondazione Cassa di Risparmio di Cuneo (CRC).

**Conflicts of Interest:** The authors declare no conflict of interest.

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
