# Peer review of "Balanced Definition of Thresholds for Mode Tracking in a Long-Term Seismic Monitoring System"

_geosciences, doi:10.3390/geosciences13120365_

Round 1

Reviewer 1 Report

Comments and Suggestions for Authors

The article is well structured.

1. Lines 146-147

How to determine for MAC each reference mode and a specific reference mode?

 2. Line 155

Formula 1 requires some explanation. The article should define MAC through the values of monitoring data and justify methods for choosing the value of the MACthr threshold. Is it unclear whether it is necessary to introduce the variable e??? in (1)?

 3. It is necessary to explain how errors were determined in modal property identification problems.

 4. Specify the year of publication in [6].

Reviewer 2 Report

Comments and Suggestions for Authors

The main task the paper – a procedure to obtain tracking of natural frequencies of structures is presented. The authors have done a lot of computational work, but there are questions about the content of the paper.

«Natural frequencies are obtained from the signals recorded by in situ sensors (e.g., accelerometers) of dynamic monitoring systems through system identification tasks [5].»

What the types of the other sensors are to register the structure natural frequencies? It is necessary to explain why accelerometers are chosen as monitoring sensors, when it is known that acceleration is a less sensitive parameter than velocity, especially in low frequencies. In addition, there are now seismometers with a large dynamic range. It is necessary to give a brief technical description of the installed accelerometers (frequency range, dynamic range, sensor sensitivity). The principle of data collection and transmission to a single data processing center should be specified. Specify where the data collection center is located. The information that the quality of transmitted data is affected by weather conditions and other adverse factors via the Internet channel is doubtful. It is known that seismometers are located in the Arctic in very unfavorable conditions, but it is possible to receive data online.

Fig. 3, 4. It is necessary to explain the color values used. The legend should be shown at the bottom of the figure and indicate which MAC values were used for each section of the figure. You need to add the letters a, b, c and so on.

Unfortunately, as we understand, the stability of the sanctuary is now threatened by progressive fractures due to aging and chemical degradation of materials, the static and dynamic effects by dead loads and ambient actions, differential settlement of foundations and so on. The question of creating of the structure computational model is very interesting. As we understand, the Sanctuary of Vicoforte is a historical pride, while it has undergone many reconstructions. An additional factor is the natural aging of materials. This means that when creating a computational model, using uniform physical and mechanical parameters for the entire structure is an incorrect solution. Such a solution leads to significant errors in determining the calculated values of the natural modes of the Sanctuary of Vicoforte vibrations.

 Fig. 6 – It is necessary to add letter designations.

Fig. 8 и Fig.  9 – It is necessary to give a general name to the figures, and then give an explanation for each part a and b.

 It is necessary to explain in the text which parameters changed when comparing the observed and calculated natural frequencies of the structure?

Reviewer 3 Report

Comments and Suggestions for Authors

The paper is interesting to the reader. It can be a good contribution to earthquake risk assessment. 

Author Response

The authors thank the reviewer very much for his time dedicated to reviewing the paper.